# Characterization of *Zanthoxylum rhoifolium* (Sapindales: Rutaceae) Essential Oil Nanospheres and Insecticidal Effects to *Bemisia tabaci* (Sternorrhyncha: Aleyrodidae)

**DOI:** 10.3390/plants11091135

**Published:** 2022-04-22

**Authors:** Karla de Castro Pereira, Eliane Dias Quintela, Vinicius A. do Nascimento, Daniel José da Silva, Dannilo V. M. Rocha, José Francisco A. Silva, Steven P. Arthurs, Moacir Rossi Forim, Fabiano Guimarães Silva, Cristiane de Melo Cazal

**Affiliations:** 1Instituto Federal de Educação, Ciência e Tecnologia Goiano—Campus Rio Verde, Rod. Sul Goiana Km 01, Rio Verde 75901-970, GO, Brazil; karla.castro@ifgoiano.edu.br (K.d.C.P.); fabiano.silva@ifgoiano.edu.br (F.G.S.); 2Empresa Brasileira de Pesquisa Agropecuária—Embrapa Arroz e Feijão, Rodovia GO-462, Km 12, Fazenda Capivara, Zona Rural, CP 179, Santo Antônio de Goiás 75375-000, GO, Brazil; eliane.quintela@embrapa.br (E.D.Q.); dannilovono@hotmail.com (D.V.M.R.); jose.arruda-silva@embrapa.br (J.F.A.S.); 3Instituto Federal de Educação, Ciência e Tecnologia Sudeste de Minas Gerais—Campus Barbacena, Rua Monsenhor José Augusto, n 204, Bairro São José, Barbacena 36205-018, MG, Brazil; viniciusnasciment@yahoo.com.br (V.A.d.N.); danieljose95@ymail.com (D.J.d.S.); 4BioBee USA, Tucker, GA 30084, USA; steven.arthurs@biobee.us; 5Department of Chemistry, Federal University of São Carlos, Rod. Washington Luiz, Km 235, CP 676, São Carlos 13565-905, SP, Brazil; mrforim@gmail.com

**Keywords:** natural insecticide, whitefly, nanotechnology

## Abstract

Encapsulation via nanotechnology offers a potential method to overcome limited thermal and photo-stability of botanical pesticides. In this study, nanospheres of essential oils (NSEO) derived from *Zanthoxylum rhoifolium* Lam. fruit were characterized and evaluated for their photostability and insecticidal activity against *Bemisia tabaci.* Three major compounds of *Z. rhoifolium* fruits were detected by CG-MS: β-phellandrene (76.8%), β-myrcene (9.6%), and germacrene D (8.3%). The nanoprecipitation method was used to obtain homogeneous spherical NSEO, with ≥98% encapsulation efficiency. Tests with UV/Vis spectrophotometry showed significantly reduced photodegradation from exposed NSEO samples when compared with essential oil (EO) controls. Whitefly screenhouses bioassays with bean plants treated with 0.25, 0.5, 1 and 1.5% suspensions showed EO treatments in both free and nanoencapsulated forms reduced adult whitefly oviposition by up to 71%. In further tests, applications at 1.5% caused ≥64% mortality of second instar nymphs. When the test was conducted under high temperature and light radiation conditions, the insecticidal effect of NSEO treatments was improved (i.e., 84.3% mortality) when compared to the free form (64.8%). Our results indicate the insecticidal potential of EO-derived from *Z. rhoifolium* fruits with further formulation as nanospheres providing greater photostability and enhanced insecticidal activity against *B. tabaci* under adverse environmental conditions.

## 1. Introduction

While global food security relies on the use of synthetic pesticides [1], there is clear evidence that overreliance on this approach has negative environmental and human health consequences [2,3]. Therefore, there is interest in developing pesticide based on natural products, which have fewer environmental costs compared with many synthetic pesticides.

Plants produce a range of secondary compounds to protect themselves from herbivores [4]. These allelochemicals are often biosynthesized through complex metabolic processes, which have evolved as both ‘static’ and ‘induced’ biochemical defenses against a range of insects [5]. For example, certain aromatic plants produce a range of ‘essential oils’, which are volatile lipophilic molecules, such as terpenes and terpenoids, phenol-derived aromatic components and aliphatic components, that have low molecular weight [6]. The components of essential oils have insecticidal properties. These include disruption of cuticular waxes and membranes, leading to desiccation, and disrupting the action of digestive and neurological enzymes of insects [7].

The silverleaf whitefly, *Bemisia tabaci* Biotype B Genn. (Hemiptera: Aleyrodidae) is a global pest of several cultivated plants [8]. The transmission of several plant viruses increases the economic impact of *B. tabaci* [9,10]. Research indicates the potential of essential oils to manage whiteflies. For example, essential oils from three different plant families exhibited both fumigant and contact toxicity against *B. tabaci*, as well as deterring oviposition [11]. In another study, Emilie et al. [12] reported that at least seven essential oils had repellent, irritant and/or toxic effects against *B. tabaci* adults. Despite these properties, the application of essential oils as insecticides is limited as they are susceptible to conversion and degradation reactions, including photodegradation [13], thermal degradation [14] and oxidative and polymerization processes [15].

Previous research has identified nanotechnology as a tool to extend the persistence and delivery parameters of certain pesticides and fertilizers [16,17]. We previously demonstrated that essential oils derived from fruits of *Zanthoxylum* spp. (Sapindales: Rutaceae) plants native to the Cerrado savannah region in Brazil exhibited repellent effect against *B. tabaci* egg-laying [18]. We also demonstrated that such essential oils from leaves could be formulated into nanoparticles [19]. It was determined that these ‘biodegradable nanospheres’ substantially protected the essential oils of *Z. riedelianum* [20] and another plant *Xylopia* spp. [21] from environmental degradation.

This study advances on previous research that tested *Z. rhoifolium* extracted from leaves [19]. Here, we extend the evaluation of *Z. rhoifolium* components extracted from plant’s fruits, which were determined to yield significantly higher concentrations of essential oils compared with leaves. Our objectives were to produce and characterize polycaprolactone (PCL) biopolymer nanospheres containing the essential oil of *Z. rhoifolium* fruits as well as to evaluate the benefits of its stability and its insecticidal and deterrent effects against *Bemisia tabaci*. PCL is an aliphatic and semi-crystalline polyester that has important characteristics for agriculture, as it has biocompatibility, low toxicity to mammals, mechanical and kinetic properties of degradation, ease of molding and manufacture of pores with appropriate sizes that allow controlling the release of active substances present in your matrix [22].

## 2. Results

### 2.1. Quantification of Z. rhoifolium Fruits

The essential oil of *Z. rhoifolium* fruits obtained by hydrodistillation gave an average yield of 0.46 ± 0.02% *w*/*w*. The GC-MS analysis revealed that β-phellandrene (76.8%), β-myrcene (9.6%), and germacrene D (8.3%) were the major compounds (Table 1).

### 2.2. Validation of UV-VIS Spectroscopy

The regression for *Z. rhoifolium* fruit essential oil concentration was described by the equation y = 5.8107x − 0.0151, where y is the mean absorbance, and x is the concentration of essential oil (mg/mL). The correlation coefficient (R^2^) was 0.9994 and adjustment was linear (*n* = 3) in the working range.

The values obtained for intraday (*n* = 3) and interday (*n* = 9) accuracy were >0.6, demonstrating that the method was highly accurate (Table 2). The mean value obtained from the accuracy was 101.5 ± 1.47, indicating a high agreement between the obtained values and the nominal values.

The limit of detection (LD) and limit of quantification (LQ) of the essential oil of *Z. rhoifolium* obtained were 0.0073 mg/mL and 0.0220 mg/mL, respectively. The LQ was lower than the first point of the curve (≤0.025). Therefore, the method used was linear and accurate in this working range.

### 2.3. Physicochemical Characterization of Nanosphere Suspensions (NSEO)

All nanoformulations showed particles values between 120 and 146 nm, i.e., within the nanometer scale (10–1000 nm). Nanoparticles containing essential oils were statistically similar in particle diameter values, polydispersity index, and zeta potential, when compared with empty nanospheres (Table 3). The pH values of the nanoformulations remained at approximately 6.0, except for the highest concentration (NS4) with a pH of 4.7. Nanospheres with essential oils from leaves of the *Z. rhoifolium* had pH values between 4 and 5. The solvent displacement method proved to be efficient for the encapsulation of the essential oil from fruits this same plant for all formulations, with values above 98% (Table 3).

### 2.4. Morphology of the Nanospheres

Scanning electron microscopic (SEM) showed spherical nanoparticles with regular surface and shape (Figure 1a). In addition, suspension homogeneity and low size dispersion of NS could be observed, corroborating the PdI values.

### 2.5. UV Protection via Nanoencapsulation

The NSEO formulation slowed the UV-degradation observed from non-encapsulated formulations. After 9 h, the EO degraded 89.2%, while the NSEO degraded 38.6% (Figure 1b), suggesting that nanotechnology protected the essential oil against photodegradation.

### 2.6. Whitefly Bioassay on Female Oviposition

In free-choice tests, adult whiteflies preferentially oviposited on plants without essential oils (Table 4). Bean leaves receiving EO and NSEO applications were less preferred suggesting a deterrent effect, with the exception of the lowest tested concentration at 0.25%. Spirofesifen also reduced oviposition rates in choice tests. There was some evidence for a repellency dose–response effect on EO and NSEO formulations, based on oviposition curves fitted by the logistic model (R^2^ = 57.76%) (Figure 2a). In no-choice tests, EO and NSEO treatments also reduced oviposition when compared to the controls, with the exception of NSEO at 0.5%. The reduction in oviposition from EO and NSEO treatment concentrations was also fitted to the logistic model (R^2^ = 66.01%) (Figure 2b). In our studies, no phytotoxicity were observed on beans leaves by either encapsulated or non-encapsulated forms of essential oils obtained from the *Z. rhoifolium* fruits.

### 2.7. Whitefly Bioassay on Nymphal Mortality

Whiteflies nymphs treated with both EO and NSEO formulations had significantly higher mortality in all three experiments, conducted at different times of the year and under different environmental conditions, when compared to water controls (Table 5). However, this mortality effect was less pronounced when compared with plants treated with Tween 80, suggesting some mortality occurred with this spreader sticker. The insecticide treatment cyantraniliprole was caused the highest mortality, killing all nymphs in the study. Mean comparisons (Table 5) suggest a mortality-concentration effect with both EO and NSEO formulations. This observation was supported by the logistic model fitted (Figure 3). In the experiment one, curves best fitted to the Gompertz model (R^2^ = 86.44%). The curves of experiments 2 and 3 were better fitted in the Weibul model (R^2^ = 68.74% and R^2^ = 85.43%, respectively). When the curves in experiment 1 were compared, the EO killed significantly more 2nd-instar nymphs than did the NSEO (*p* = 0.021) (Figure 3a). In the second experiment, the insecticidal effect of NSEO of *Z. rhoifolium* was more significant than that of EO at the lowest and highest concentrations (*p* = 0.047) (Figure 3b). In the third experiment, no significant difference was observed in the mortality of nymphs between the doses 1.0% and 1.5% of EO and NSEO (*p* = 0.06207) (Figure 3c), respectively. The highest doses (1.5%) killed more than 76% of the *B. tabaci* nymphs.

## 3. Discussion

In the present study, we prepared polycaprolactone (PCL) biopolymer nanospheres containing essential oil of *Z. rhoifolium* fruits (NSEO). These nanoparticles were spherical, with dimensions between 120 and 146 nm, and formed a homogenous suspension in water. Our results showed that the essential oil from *Z. rhoifolium* fruits had high extraction yield, excellent encapsulation rate in PCL, in addition to promising activity in the control of *B. tabaci*.

The size and homogeneity of nanoparticles containing pesticides are important factors predicting their properties [16,17]. The size and polydispersity of nanoparticles obtained in our study validates that a monodispersion was obtained [23,24]. Another parameter in determining the stability of colloidal suspensions is ZP (ζ) which quantifies the electric potential of the particles (electrostatic repulsion). High values of ZP (ζ > 30 mV), indicate physicochemical stability, since the charged particles repel each other avoiding aggregation [25,26]. In our samples, all ZPs were >20.2 mV and can be considered stable.

Our solvent displacement method proved efficient for encapsulation of *Z. rhoifolium* fruits essential oils, with values above 98%. These values demonstrate the affinity of essential oils to the PCL polymer matrix. Using a slightly different method, Pinto et al. [27] reported an EE% of 70.6% in nanoformulations containing 2.5% of essential oil of *Lippia sidoides* leaves. Nanocapsules of chitosan and alginate containing essential oil of saffron and lemon grass gave an EE of 71.1% and 86.9%, respectively [28]. Jamil et al. [29] obtained an EE% above 90% for nanoparticles of chitosan containing cardamom essential oil.

Photodegradation limits the longevity of natural and synthetic pesticides [30]. We showed that nanotechnology helps protect the essential oil of *Z. rhoifolium* fruits against this process. Previous studies also reported nanospheres provided UV-protection for essential oil extracts from *Z. rhoifolium* leaves [19] and *Z. riedelianum* fruits [20]. Our study confirms the value of nanoparticle encapsulation for photoprotection of certain botanical compounds.

An important aspect of this study was to quantify the insecticidal potential of *Z. rhoifolium* fruits against *B. tabaci*, which is a global pest. Bean leaves treated with EO and NSEO exhibited a deterrent to whitefly oviposition. Our results confirm that exposure of adult *B. tabaci* to *Z. rhoifolium* fruits extracts reduced oviposition by more than 70%. The nanoencapsulated formulations of *Z. rhoifolium* showed better deterrent activity than *Z. riedelianum* in the no-choice test [20]. This result can be explained by the environmental conditions during the experiment, since both species have deterrent potential. Moreover, the ability to formulate these treatments in nanospheres may be expected to enhance the longevity of such effects under field conditions. Future research is needed to assess the longevity of the nanoencapsulated essential oil formulations in the field.

The oviposition deterrent activity may be explained by several factors. First, the lipiophilic nature of the essential oil may prevent egg fixation on the host plant tissue [31]. There is also evidence that volatile compounds from essential oils may impact ovary development and egg maturation [32,33]. Another possible factor for reduced oviposition is that adult phloem-feeding might be disrupted by the tactile and volatiles cues on the leaf surfaces [31,34].

In addition to sub-lethal effects on adults, our data show that both EO and NSEO treatments killed second instar *B. tabaci*, with up to 91% mortality in one experiment. There was some evidence that NSEO treatments were more effective in some cases. These results expand on a previous study which indicated insecticidal activity of nanoencapsulated *Z. riedelianum* fruits, where a 1.5% concentration killed >80% of the second instar *B. tabaci* [20]. In both species, the insecticidal activity was better at higher concentrations, which may represent a promising genus in phytosanitary defense.

The deterrent and insecticidal activity of the essential oils of *Z. rhoifolium* fruits may be associated with their major compounds. In the current study, these comprised β-phellandrene (76.8%), β-myrcene (9.6%), and germacrene D (8.3%). Prieto et al. [35] identified similar compounds in fruits of three Colombian *Zanthoxylum* species, but at different proportions, i.e., β-myrcene (59.0%), β-phellandrene (21.5%), germacrene D (9.3%), bicyclogermacrene (3.1%), and 2-undecanone (1.7%). Costa et al. [18] also found β-myrcene (8.0%) and germacrene D (17.1%) in another *Zanthoxylum* spp., but the compound with the highest amount was sabinene (55.9%). De Gonzaga et al. [36] identified only four components in the essential oils of fruits of *Z. rhoifolium* collected in Rio Grande do Sul state, Brazil, among them: ment-2-en-1-ol (46.2%), β-myrcene (30.2%), (-)-linalool (15.0%), and (-)-α-terpineol (8.5%). The variation in the composition of essential oil compounds noted above may relate to several factors, including plant species/cultivar, stage of plant development, growing conditions, and nutrition [37].

Previous research also confirms direct insecticidal effects of the same volatile EO constituents against other insect pests. Seven volatiles, including myrcene and α-phellandrene isolated from leaves of the hinoki cypress tree, *Chamaecyparis obtuse*, were determined to be insecticidal to two stored product beetles. In impregnated paper bioassays, treatment of 0.1 mg volatile/cm^2^, killed up to 97% of exposed *Callosobruchus chinensis* and 93% of *Sitophilus oryzae* within 2 days [38].

The direct insecticidal action of EO may have multiple causes, which require further investigation. Oils can interfere with insect cuticle function, including respiration and water regulation activity [39]. Since EO are composed of a complex of substances, including terpene, esters, aldehydes, and alcohols, their action may further reflect synergistic or additive effect of different active compounds, even if some compounds act in isolation [6]. Some EO compounds, such as monoterpenes and sesquiterpenes, may cause neurotoxicity by interfering with the octopamine neuromodulator or the GABA-gated chloride channels [40]. The combination of substances may reduce the selection pressure for resistance of insect pests [12,40].

As a final note, our screenhouse studies conducted at different times of the year suggest that treatments may depend upon the prevailing environmental conditions (Appendix A). In particular, applications of some nanoformulated treatments during higher temperature (up to 33 °C) (experiment 2) and sunlight intensity resulted in higher nymphal mortality compared with unformulated EO (experiment 1 and 3). This variation suggests that the climatic factors may accelerate the degradation of the active compounds. For example, the cleavage of the nano-polymers may occur more quickly under warmer conditions, causing a more rapid release of insecticidal molecules, but also potentially reducing their longevity. Carvalho et al. [41] attributed the differences in insecticidal activities of two formulations of neem oil nanocapsules (NC L5-2 and NC L6-1) to environmental factors.

## 4. Materials and Methods

### 4.1. Essential Oil Extraction

Fruits of *Z. rhoifolium* were collected in the municipality of Hidrolândia-GO (17°00′52.1″ S and 49°12′3.5″ W) and Iporá (16°26′44.5″ S and 51°7′58.7″ W). Fruits with seeds showing the first signs of carotenoid accumulation on external surface (pre-ripening stage) were used for extraction by hydrodistillation for a period of 3 h using a Clevenger-type apparatus coupled with a 3 L round bottom flask, as described previously [20]. Essential oils extracts were pooled from different collection points to obtain sufficient material for bioassays and CG-MS analysis. The samples were transferred to a 20 mL amber bottle and refrigerated the oil yield (%) was determined by the ratio between the weight (g) of the oil obtained and the weight of fruit (500 g) used in the extraction.

### 4.2. CG-MS Analysis

Qualitative and quantitative analyses of the *Z. rhoifolium* essential oil constituents were conducted via gas chromatograph-mass spectrometry (GC-MS). Samples were processed via a Combi PAL AOC-5000 auto-injector (CTC Analytics, Aargau, Switzerland), Restek Rtx-5ms fused silica capillary column (30 m × 0.25 mm × 0.25 µm) (Restek, Bellefonte, PA, USA), a sequential mass spectrometer (MSTQ8030, Shimadzu, Tokyo, Japan), and an ionization detector of electron impact (IE) (70 eV). The temperature profiles were 60 °C for 3 min, rising to 200 °C (at 3 °C min^−1^), and then 280 °C (at 15 °C min^−1^) and held for 1 min. The injector and detector temperature were 230 °C and 300 °C, respectively. The carrier gas was helium with injection pressure of 57.4 KPa, in the splitless mode: 150, mass detection range from 43 to 550 Da, and flow rate of 3 mL.min. Results were analyzed using CG-MS Real Time Analysis^®^ software.

### 4.3. Spectrophotometry

Quantification of the *Z. rhoifolium* essential oil was assessed on a spectrophotometer (DR/5000 UV-Vis HACH, Hach Company, Loveland, CO, USA) with absorbance at 248 nm. The calibration curve was based on mean absorbance values in the UV/Vis region at the following concentrations: 0.025, 0.05, 0.1, 0.15, 0.2, and 0.25 mg/mL. Three concentrations covering the working range were prepared, i.e., 0.030, 0.125, and 0.225 mg/mL. Results from these different concentrations were tested on three independent samples at the same time (intraday) and also on different days (interday).

The limit of detection (LD) and the limit of quantification (LQ) of *Z. rhoifolium* EO samples were calculated from the standard deviation of the intercept (0.013) and the slope (5.759) of the absorbance calibration curve (0.025, 0.05, 0.1, 0.15, 0.2, and 0.25 mg/mL):$$\mathrm{DL}=\frac{3\;*\;s}{S}$$
$$QL=\frac{10\;*\;s}{S}$$
where *s* is the standard deviation of response, and *S* is the calibration graph coefficient.

### 4.4. Essential Oil Nanoparticle Preparation and Characterization

Nanoparticles (NP) were prepared using the preformed polymer nanoprecipitation method [42]. The organic phase was prepared at 40 ± 3 °C and under stirring, with the suspensions prepared from 150 mg of PCL biopolymer, 60 mg of Span^®^ 60 (low-hydrophilic-lipophilic balance [HLB] surfactant), 10 mL of acetone PA (organic solvent), and different amounts (mg) of essential oil. The organic phase was decanted over an aqueous phase containing 20 mL of distilled water and 50 mg of Tween^®^ 80 (high HLB surfactant) using a peristaltic pump (Watson-Marlow) while stirring for 10 min for stabilization. The water and organic solvent were eliminated via a rotary evaporator, to the final volume of colloidal suspension.

Four formulations NS 1, NS 2, NS 3 and NS 4 containing 0, 50, 100 and 250 mg of essential oil, respectively. Suspension pHs were measured with a potentiometer (HANNA model FT-P21, Tecnal Scientific Equipment, Piracicaba, SP, Brazil). To assess the stability (zeta potential) of suspensions, the surface charge of the particles and particle diameter (SD) in suspension were measured with a ZetaSizer Nano Z-S instrument (Malvern Instruments, Malvern, UK). Samples were diluted with distilled water to give a final concentration of 10% *v/v* and analyzed in triplicate. Data were analyzed by one-way ANOVA, and the means were compared by the Tukey’s test (*p* ≤ 0.05).

### 4.5. Encapsulation Efficiency

The amount of encapsulated essential oil was determined using the filtration–centrifugation method. Aliquots of 0.5 mL of the nanoparticle colloidal suspensions were transferred to tubes with 0.22 µm pore cellulose acetate filters (Spin-X, Corning^®^ Inc., Corning, NY, USA) and subjected to refrigerated centrifugation (Thoth 9300R, Thoth Equipment, Piracicaba, Brazil) at 8000 rpm and 20 °C for 40 min. Subsequently, 250 μL of the ultrafiltrate was removed and added to 2.5 mL of hexane. The hexane fraction was separated by liquid–liquid extraction using a vortex and analyzed by UV/Vis spectroscopy.

Encapsulation efficiency (% EE), was calculated by the difference between the total amount of essential oil used in the sample preparation and the total amount of essential oil in the ultrafiltrate with the equation:$$EE\%=\frac{B-A}{B}\;*\;100$$
where *A* is the total concentration of essential oil in the ultrafiltrate (mg/mL), and *B* is the total concentration of essential oil in the suspension (mg/mL).

### 4.6. Nanoparticle Morphology

The homogeneity of the colloidal suspension and shape of *Z. rhoifolium* essential-oil- nanoparticles were evaluated at the Microscopy Center of the Federal University of Minas Gerais (UFMG). Samples of the colloidal suspension was deposited onto 12 mm diameter aluminum ‘stubs’ and dried in silica. After solvent evaporation, samples were sputtered with 2 nm of Au/Pd alloy and analyzed by Scanning Electron Microscopy (MEV-Quanta 200 FEI, Thermo Fisher Scientific, Waltham, MA, USA).

### 4.7. UV Light-Accelerated Degradation

UV light-accelerated degradation of non-encapsulated and nanoencapsulated essential oils were analyzed with an ultraviolet chamber (BOIT-LUV01, BOITTON Instrumentos, Porto Alegre, Brazil) containing a UV-A and UV-B radiation lamps providing 365 nm and 254 nm wavelengths, respectively. Observations were conducted at 25.0 ± 2.0 °C. Aliquots of 1 mL per sample were transferred to clear vials and exposed to the UV chamber for 0, 1, 2, 3, 5, 7 and 9 h exposure intervals. Subsequently, 0.25 mL of each exposed sample was diluted with 1.5 mL of hexane. The solvent and essential oil were then separated by vortexing and analyzed by UV/Vis spectrophotometry to assess essential oil photostability.

### 4.8. Bioassays with B. tabaci

Insect trials were conducted in a 50-mesh screened house (18 × 7 × 4 m high) located at Embrapa Rice and Beans, Santo Antônio de Goiás, GO, Brazil. Temperature, relative humidity and light intensity were monitored via datalogger (Hobo^®^ U12-012, Onset Computer Corp. Ltd., Bourne, MA, USA). Adult whiteflies were obtained from a colony reared on common beans (*Phaseolus vulgaris* L., cv. Pérola) and identified as *B. tabaci* Biotype B by molecular gene sequence markers from mtDNA cytochrome oxidase I (mtCOI) [43]. Bean plants were grown in plastic pots containing 0.4 or 2 L of soil (Eutrophic Red Latosol) for the experiments with adults and nymphs, respectively.

#### 4.8.1. Oviposition Deterrent Activity against *B. tabaci*

The deterrent activity of *Z. rhoifolium* essential oil-loaded nanoparticles (NSEO) against whitefly oviposition behavior were evaluated at four concentrations (0.25%, 0.5%, 1%, and 1.5%). In addition, distilled water, suspensions of empty nanoparticles (excluding essential oil), and the chemical insecticide Oberon^®^ (spiromesifen, Bayer AG, Dormagen, Germany) at 0.25% and an emulsifier/spreader (0.3% Tween^®^ 80) were used for comparison.

Both choice and no choice tests were performed, with four replications per treatment in a completely randomized experimental design. Each replicate consisted of one common bean seedling (12 days old) containing two primary leaves and grown in 2 L pot. Both surfaces of each primary leaf were sprayed with 300 µL of each treatment using an airbrush sprayer (Paasche H-set) coupled to a vacuum pump. The climatic conditions during screenhouse experiments were 29.1 ± 1.7 °C; 44.3 ± 5.6% RH and 8049.2 ± 794.4 lm/m^2^ light intensity.

In choice tests, whiteflies were provided plants from all treatment (48 plants, four repetition per treatment). Plants were placed inside cages (100 × 45 × 45 cm high) that were screened with voile fabric (12 plants per cage). Each choice cage received 350 adult whiteflies that could oviposit for 24 h. Subsequently, leaves and insects were removed (2 leaves/repetition, 8 leaves per treatment). The number of eggs in the middle of the leaf abaxial surface (4 cm^2^ area) were counting at 40× with a stereomicroscope Leica EZ4 (Leica Microsytems, Switzerland, Ltd., Heerbrugg, Switzerland).

In no-choice test, each cage (30 × 30 × 50 cm high) received one pot containing one common bean seedling with two primary leaves. After treatments application, 30 adult whiteflies were released per cage for 24 h for oviposition, and leaves removed for evaluation, as described above. Each treatment had four repetitions. For the assessments, the numbers of eggs laid were verified by the Kolmogorov–Smirnov test and homogeneity of variances by the Levene’s test. Treatments were analyzed by ANOVA with egg numbers and oviposition indices compared by Kruskal–Wallis and *t* tests, respectively. A stimulant/deterrent oviposition index was also calculated using the formula proposed by Fenemore (1980) [44] [(A − B)/(A + B)] × 100, where A = the number of eggs in the treatment, and B = the number of eggs in the control. The index ranged from +100 (total stimulation), through zero (neutral), to −100 (total deterrence).

For both choice and no choice experiments, dose–response curves were fitted for EO and NSEO using the 4-parameter generalized log-logistic model for a binomial response:$$y=c+\frac{d-c}{1+\exp\left(b\left(x-e\right)\right)}$$
where *y* is the number of eggs, *b* the slope of the dose–response curve, *c* is the lower limit, *d* is the upper limit, and *e* is the effective dose ED50.

#### 4.8.2. Mortality of *B. tabaci* Nymphs

Additional experiments evaluated the effects of NSEO *Z. rhoifolium* NSEO at 0.25, 0.5, 1 and 1.5% on 2nd instar whiteflies. Distilled water, suspensions of empty nanoparticles (excluding essential oil), and the anthranilic diamide insecticide cyantraniliprole in the form of an oil dispersion (Benevia^®^, FMC Corporation, Campinas, SP, Brazil) at 0.25% were used for comparison.

Pest-free common bean plants (10 days old with two primary leaves) were exposed to a controlled infestation of adult-whiteflies for an eight-hour oviposition period. This procedure provided ≈100 eggs per leaf. The newly infested plants moved to another screenhouse until nymphs reached the 2nd instar.

Treatments were applied to the abaxial side of primary leaves containing nymphs with a microsprayer (0.3 mm needle, Paasche^®^ airbrush type H-set) connected to a vacuum pump and calibrated to 250 μL per leaf. The mortality of nymphs was assessed in the middle of the leaf abaxial surface (4 cm^2^ area) with a 40× stereomicroscope. Dead nymphs were determined based on the brown color and desiccated aspect. Mortality evaluations were conducted on four leaves per treatment starting on day 3 after spraying and continued on alternate days for eleven days. Empty pupa cases were used to determine the survival of the 4th instar nymphs. The experiment was arranged in a completely randomized design, with four replications per treatment and three plants per repetition.

For analysis, nymphal mortality (%) data was tested by the Kolmogorov–Smirnov test to verify the residual normality, and homogeneity of the data was verified by the Levene’s test. All data were analyzed by ANOVA and means were compared by the Scott Knott test (*p* ≤ 0.05).

The experiment was conducted 3 times using independent samples. In the first experiment, dose–response curves were constructed for EO and NSEO using the 3-parameter generalized Gompertz model for binomial response data:y=(d)(exp(−exp(b(x−e))))

For experiments 2 and 3, dose–response curves were constructed for EO and NSEO using the generalized 3-parameter Weibul model for binomial response data:Experiment 2: y=c+(1−c)(1−exp(−exp(b(log(x)−log(e)))))
Experiment 3: y=c+(1−c)(exp(−exp(b(log(x)−log(e)))))
where *y* is proportional mortality, *b* the slope of the dose–response curve, *c* is the lower limit, *d* is the upper limit and is the LC50.

Dose–response curves were compared for parallelism between EO and NSEO using Wilcoxon’s non-parametric test at *p* < 0.05. The software application used to analyze the data was R version 3.1.2 [45].

## 5. Conclusions

In summary, we demonstrated the potential of nanoencapsulation to enhance the direct and indirect insecticidal effects of essential oils. The nanotechnology potentiated the effects of essential oils by reducing the photodegradation of their active compounds. Scaled-up field trials with different agricultural pests and further research on the interactions of the nanosphere technology with different pesticides and various environmental factors are needed to better validate the potential of this approach.

## Figures and Tables

**Figure 1 plants-11-01135-f001:**
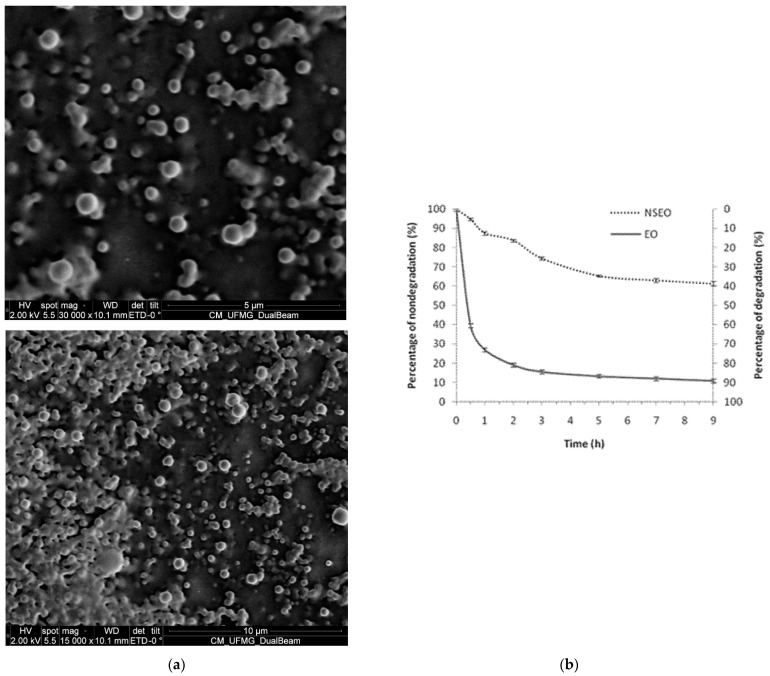
(**a**) Scanning electron microscope (SEM) of suspensions of PCL nanospheres containing the essential oil of *Z. rhoifolium* fruits. (**b**) UV-accelerated degradation of the essential oil in the free (EO) and nanoencapsulated forms (NSEO).

**Figure 2 plants-11-01135-f002:**
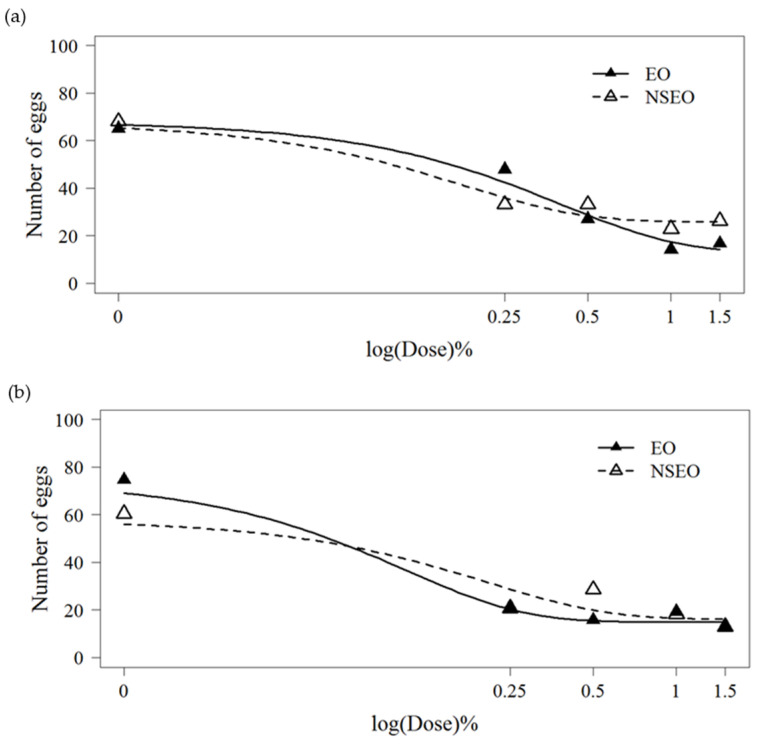
Mean number of *B. tabaci* eggs on bean leaves treated with different concentrations of the essential oil of *Z. rhoifolium* fruits in free (EO) and nanoencapsulated (NSEO) forms in the free-choice test (**a**) and the no-choice test (**b**). Dose–response curves were fitted using the 4-parameter generalized adjusted according to log-logistic model for a binomial response data.

**Figure 3 plants-11-01135-f003:**
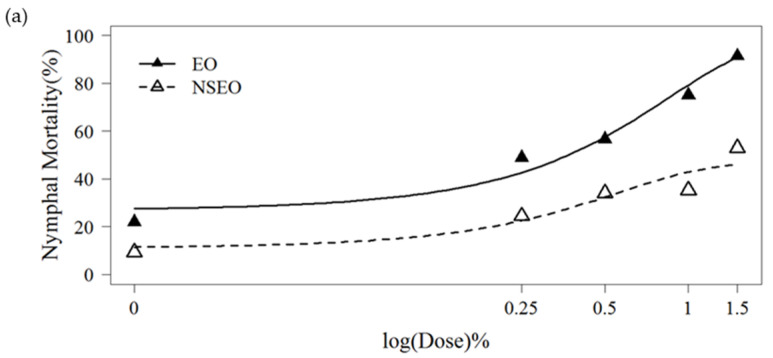
Mortality of 2nd-instar nymphs of *B. tabaci* after application of essential oil in the free (EO) and nanoencapsulated (NSEO) forms at different concentrations for experiment 1 (**a**), experiment 2 (**b**), experiment 3 (**c**). Curves were adjusted according to 3-parameter generalized Gompertz model for binomial response data (**a**) and Weibull (**b**,**c**).

**Table 1 plants-11-01135-t001:** Chemical composition of the essential oil of *Z. rhoifolium* fruits analyzed by gas chromatography-mass spectrometry (GC-MS).

Peak	TR (min)	Compounds ^a^	Mass (%) ^a^	RI Exp. ^b^	RI Lit. ^c^
1	7.240	β-myrcene	9.59	992	990
2	8.519	β-phellandrene	76.77	1030	1029
3	11.051	Linalool	0.58	1101	1096
4	14.648	Cryptone	0.88	1189	1185
5	19.128	2-undecanone	0.50	1295	1294
6	20.993	δ-elemene	0.28	1340	1338
7	22.615	α-copaene	0.49	1379	1376
8	22.930	Geranyl acetate	0.18	1386	1381
9	24.408	Caryophyllene	0.77	1422	1419
10	26.735	γ-muurolene	0.23	1480	1479
11	26.915	Germacrene D	8.34	1484	1485
12	27.553	Bicyclogermacrene	0.49	1500	1500
13	28.605	α-cadinene or naphthalene	0.63	1527	1523
14	30.706	Spathulenol	0.27	1582	1578

^a^ GC-MS analyses. ^b^ Experimental retention index. ^c^ Literature retention index.

**Table 2 plants-11-01135-t002:** Precision (RSD%) and accuracy (%) of essential oil samples of *Z. rhoifolium* fruits used in the validation of the analytical method.

	RSD %				Accuracy
Concentration (mg/mL)	Intraday 1	Intraday 2	Intraday 3	Interday	Interday (%)
	(*n* = 3)	(*n* = 3)	(*n* = 3)	(*n =* 9)	(*n =* 9)
0.03	0.3 ± 0.001	0.3 ± 0.001	0.6 ± 0.001	0.4 ± 0.2	103.5 ± 0.2
0.125	0.2 ± 0.002	0.1 ± 0.001	0.2 ± 0.002	0.2 ± 0.1	101.0 ± 0.2
0.225	0.1 ± 0.001	0.0 ± 0.001	0.1 ± 0.001	0.1 ± 0.1	100.0 ± 0.2

**Table 3 plants-11-01135-t003:** Particle diameter (PD) values, polydispersity index (PdI), zeta potential (ZP), pH and percent encapsulation efficiency (EE%) of nanosphere suspensions (NS) containing essential oil of *Z. rhoifolium* fruits.

Formulations	Essential Oil (mg)	PD (nm) *	PdI *	ZP (mV) *	pH *	EE %	Number of Samples
NS 1	0	120.5 ± 8.77 ^a^	0.22 ± 0.01 ^a^	−24.1 ± 5.8 ^a^	6.64 ± 0.04 ^a^		3
NS 2	50	129.7 ± 3.12 ^a^	0.24 ± 0.02 ^a^	−20.2 ± 3.4 ^a^	6.35 ± 0.02 ^a^	98.04 ± 0.04	3
NS 3	100	137.1 ± 12.43 ^a^	0.24 ± 0.03 ^a^	−24.9 ±1.1 ^a^	6.18 ± 0.37 ^a^	98.57 ± 0.07	3
NS 4	250	145.53 ± 14.23 ^a^	0.25 ± 0.03 ^a^	−22.1 ±1.0 ^a^	4.77 ± 0.11 ^b^	99.46 ± 0.11	3

* Averages with the same letters in the same column indicate that there was no significant difference, according to Tukey’s test (*p* < 0.05).

**Table 4 plants-11-01135-t004:** Free-choice and no-choice tests comparing *B. tabaci* oviposition after treatment with essential oil extracts in the free (EO) and nanoencapsulated (NSEO) form of *Z. rhoifolium* fruits.

Treatments	Doses (%)	Eggs ^1^	Oviposition Index ^2^ (%)
Free-choice test			
EO	0.25	47.9 ± 24.6 abc	−15.3 ns
	0.5	27.0 ± 15.5 cd	−41.4 **
	1.0	14.2 ± 12.1 d	−64.1 **
	1.5	16.7 ± 11.7 d	−59.1 **
NSEO	0.25	33.2 ± 17.0 cd	−34.4 **
	0.5	33.2 ± 15.8 cd	−34.4 **
	1.0	22.9 ± 7.1 cd	−49.7 **
	1.5	26.4 ± 7.6 cd	−44.2 **
Water Control	-	79.5 ± 26.6 a	-
Tween^®^ 80	0.3	65.1 ± 10.5 ab	-
NS Control	-	68.1 ± 19.4 ab	-
Spiromesifen	0.25	26.6 ± 16.7 cd	−49.8 **
**No-choice test**			
EO	0.25	20.2 ± 10.1 c	−57.3 **
	0.5	15.7 ± 9.6 c	−65.1 **
	1.0	19.2 ± 8.9 c	−59.0 **
	1.5	12.7 ± 10.9 c	−70.8 **
NSEO	0.25	20.7 ± 13.2 c	−48.9 **
	0.5	28.6 ± 13.6 bc	−35.8 **
	1.0	18.4 ± 12.0 c	−53.4 **
	1.5	13.0 ± 9.8 c	−64.6 **
Water Control	-	84.6 ± 15.0 a	-
Tween^®^ 80	0.3	74.6 ± 20.2 ab	-
NS Control	-	60.5 ± 21.3 ab	-
Spiromesifen	0.25	21.0 ± 7.9 c	−60.2 **

^1^ Means followed by different letters are significantly different by the Kruskal–Wallis test (*p* < 0.05). ^2^ The oviposition index was calculated from the expression proposed by Fenemore et al. (1980), [(A − B)/(A + B)] × 100, where A = number of eggs in the test treatment, and B = number of eggs in the control treatment. For EO treatments, the Tween 80 treatment was used as comparison control. For NSEO treatments, the comparison control was empty nanospheres (NS control). For the insecticide control, the comparison control was water. ns—not significant. ** Significant (*p* < 0.05).

**Table 5 plants-11-01135-t005:** Mortality of 2nd-instar nymphs of *B. tabaci* after treatment of bean leaves with essential oil of *Z. rhoifolium* fruits in the free (EO) and nanoencapsulated (NSEO) form in three experiments in a screenhouse.

Treatments	Doses (%)	Experiment 1 (%) ^1^	Experiment 2 (%) ^1^	Experiment 3 (%) ^1^
EO	0.25	48.9 ± 16.8 c	45.4 ± 9.8 c	42.5 ± 8.4 c
	0.5	56.6 ± 7.0 c	50.3 ±8.0 b	60.2 ± 10.2 c
	1.0	74.9 ± 11.3 b	60.0 ± 9.7 b	68.4 ± 5.3 c
	1.5	91.3 ± 5.1 a	64.8 ± 9.7 b	77.6 ± 14.0 b
NSEO	0.25	24.5 ± 5.9 d	68.9 ± 16.2 b	26.0 ± 0.3 d
	0.5	34.0 ± 8.0 d	57.3 ± 16.3 b	26.3 ± 1.9 d
	1.0	35.4 ± 8.2 d	57.7 ± 8.2 b	58.3 ± 7.0 c
	1.5	53.0 ± 0.8 c	84.3 ± 0.8 a	76.2 ± 0.9 b
Water Control	0	0.9 ± 1.0 e	1.9 ± 1.3 d	1.2 ±0.9 e
Tween^®^ 80	0.3	22.0 ± 14.2 d	20.0 ± 3.6 c	21.1 ± 10.1 d
NS Control	0	9.4 ± 6.6 e	41.8 ± 6.1 c	21.9 ± 7.4 d
Cyantraniliprole^®^	0.25	100.0 ± 0.0 a	93.0 ± 7.4 a	100.0 ± 0.0 a

^1^ Means followed by different letters are significantly different by the Scott Knott test (*p* < 0.05).

## Data Availability

Not applicable.

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
