# Peer review of "Characterization of *Zanthoxylum rhoifolium* (Sapindales: Rutaceae) Essential Oil Nanospheres and Insecticidal Effects to *Bemisia tabaci* (Sternorrhyncha: Aleyrodidae)"

_plants, 2022, doi:10.3390/plants11091135_

Round 1
Reviewer 1 Report
The paper: Characterization of Zanthoxylum rhoifolium (Sapindales: Rutaceae) essential oil nanospheres and insecticidal effects to Bemisia tabaci (Sternorrhyncha: Aleyrodidae) by Karla Pereira, Eliane Quintela, Vinicius Nascimento, Daniel da Silva, Dannilo Rocha, José Francisco Silva, Fabiano Silva, Steven Arthurs and Cristiane Cazal contain interesting data based on the study which was a continuation of previously published experiments. The study has been well designed and conducted properly. It is well written and well presented. I do not have any major comments. However, there are two issues for clarification:
- Qualitative and quantitative analyses of the Z. rhoifolium essential oil constituents: It is well known that the composition of plant oils deriving from samples collected at different locations may differ. What samples were analysed? Did you use the pooled samples as described for bioassays (lines 284-290)?
- Please, give more details on the PCL nanoparticles in the Introduction. This would provide the reader with some knowledge on why this particular material was used without the need to reach for the previous work of the Authors (ref. 20)
Reviewer 2 Report
Bemisia tabaci is a worldwide pest. Its populations, especially under cover crops, are growing rapidly. Resistance to the insecticides used is widespread. In addition, the number of active substances authorized for use is reduced. Therefore, research into new pest control methods, especially when it comes to biological preparations, is very valuable. The manuscript presented for evaluation was prepared very carefully. No irregularities were found in the research methods and statistical analyzes. Research results and discussion are at an appropriate level. Literature sources were used following the discussed research topic. I have no comments.
Author Response
We would like to thank you for your careful review of our manuscript. We are very happy to receive these supportive comments and fully appreciate your valuable comments.
Reviewer 3 Report
Review of MS 1672118 submitted to Plants - Characterization of Zanthoxylum rhoifolium (Sapindales: Ru- 2 taceae) essential oil nanospheres and insecticidal effects to Bemisia tabaci (Sternorrhyncha: Aleyrodidae)
The MS deals with a interesting topic concerning a method to overcome limited thermal and photo-stability of a botanical pesticides by encapsulation. The topic presented by the authors is of some interest as it contributes both to improve specific knowledge on performance of a plant protection products and also highlight the role of a botanical species such as Z. rhoifolium. Overall, the MS is not enough detailed even if the approach adopted by the authors is re-proposable. An aspect to highlight in the MS is to highlight in greater detail the differences in terms of insecticide activity with a botanical species very similar to the one investigated, the Zanthoxylum riedelianum and given that the ratio between the components studied is different which aspects are to be considered with greater attention to use in phytosanitary defense.
Specific comment
Line 26: Write the full name for the first time of EO;
Line 27. The term (NSEO) seems superfluous to me since it has already been written nanoencapsulated;
Line 72-75. It would be advisable to provide some additional information (e.g. thermal stability) on the PLC in the text;
Line 78-81. In a previous MS published by te author in Molecules 2018, 23, 2052; doi: 10.3390 / molecules23082052 is the ratio of the detected molecules different ?. Zanthoxylum riedelianum and Zanthoxylum rhoifolium are the same species? How do you justify this difference? It depends on the ripening of the fruit or on other factors that are not easily identifiable..
Line 83. Z. rhoifolium in italics.
Line 119. Table 3. Add the columns showing the number n = (number of samples);
Line 155. Means were compared by the Tukey’s test and not wuth the Kruskal-Wallis test
Line 178, 180. p = 0.02056 in 0.021 and p = 0.04691 in 0.047
Line 284-289. Provide information on the amount of fruit used. Furthermore, the oils are mostly contained in the fruit and or in the seeds of the fruit;
Line 327. Add space for stirringfor;
Line 331. Why was this measurement done (pH) ??? Do you suspect it is an easily alterable oil?;
Line 378-379. Provide the accession number of the sequences produced;
Line 384-395. In this test the Bemisia tabaci was used but it is necessary to provide more information. The test was carried out in a climatic chamber due to the low thermal variation during the test. What was the light / dark hours ratio adopted?
Line 391-392. How long after sowing the seedlings were used to evaluate the oviposition;
Line 401 and 431. Indicate the model and brand of the stereoscope;
Line 405-407. Why is the Kolmogorov-Smirnov non-parametric test used? generally this test is used to compare a sample with a reference distribution or to compare two samples; if the information on the distribution of the data is obtained, it is necessary to report it in the results;.
